# Device-independent certification of indefinite causal order in the quantum switch

Tein van der Lugt [ORCID][1] ✉, Jonathan Barrett [ORCID][1,2] & Giulio Chiribella [ORCID][1,2,3] ✉

Quantum theory is compatible with scenarios in which the order of operations is indefinite. Experimental investigations of such scenarios, all of which have been based on a process known as the quantum switch, have provided demonstrations of indefinite causal order conditioned on assumptions on the devices used in the laboratory. But is a device-independent certification possible, similar to the certification of Bell nonlocality through the violation of Bell inequalities? Previous results have shown that the answer is negative if the switch is considered in isolation. Here, however, we present an inequality that can be used to device-independently certify indefinite causal order in the quantum switch in the presence of an additional spacelike-separated observer under an assumption asserting the impossibility of superluminal and retrocausal influences.

The past decade has seen increasing interest in the study of quantum processes incompatible with a well-defined order between operations, a phenomenon now known as causal nonseparability[1–7]. The archetypal example of such a process is the quantum switch[2], which applies two operations to a target system in a superposition of orders. This process has found numerous applications in information processing tasks such as channel discrimination[8], query complexity[9–11], communication complexity[12], transmission of information through noisy channels[13–20], metrology[21,22], and thermodynamics[23–25].

In recent years, a number of strategies to certify indefinite causal order in the quantum switch have been developed[5,26–28] and adopted in experimental investigations[29–35]. A common characteristic of these strategies is that they are device-dependent, in the sense that they rely on assumptions on the devices used in the laboratory and the physical theory that governs them. To provide stronger evidence of indefinite causal order, it is desirable to have a device-independent certification, which only relies on the statistics of measurement outcomes, in the same way as violation of a Bell inequality certifies Bell nonlocality.

For some causally nonseparable processes, such device-independent certification is possible through the violation of so-called causal inequalities[3,6,36–38]; however, the physicality of these processes is still unclear[39–42]. The quantum switch, on the other hand—the only causally nonseparable process to have been studied experimentally—has been shown not to violate any such inequality[5,6], a result that was recently extended to the broader class of quantum circuits with quantum control of causal order[43,44]. As a consequence, a device-independent certification of indefinite causal order for the quantum switch has so far been missing, leaving open the question whether it is compatible with a hidden variable description in which the order is well-defined.

In this paper we extend the standard causal inequality scenario by adding a spacelike-separated party. We derive a set of device-independent inequalities satisfied by all correlations observed in experiments satisfying the three assumptions of 'Definite Causal Order', 'Relativistic Causality', and 'Free Interventions', the second of which rules out causal influences outside the future lightcone. We then show that these inequalities are violated by a quantum process involving the quantum switch and an additional system entangled to the switch's control qubit. This establishes a device-independent certification of indefinite causal order for the quantum switch, under the assumptions of Relativistic Causality and Free Interventions. Crucially, our notion of Relativistic Causality is strictly weaker than Bell Locality, which is already known to be violated by quantum physics[45,46]; in particular, it (together with Free Interventions) only entails parameter independence, while Bell Locality also requires outcome independence[47]. In addition to deriving the inequalities and their violation, we begin to unravel the structure of the corresponding correlation polytope, which shares features with causal polytopes, no-signalling polytopes, and Bell-local polytopes.

[1]Department of Computer Science, University of Oxford, Wolfson Building, Parks Road, Oxford OX1 3QD, United Kingdom. [2]Perimeter Institute for Theoretical Physics, Waterloo, ON N2L 2Y5, Canada. [3]QICI Quantum Information and Computation Initiative, Department of Computer Science, The University of Hong Kong, Pokfulam Road, Hong Kong, Hong Kong. ✉e-mail: teinvdlugt@gmail.com; giulio@cs.hku.hk

## Results

### Device-independent inequality

We will consider an experiment carried out by four agents, Alice 1 ($\mathcal{A}_1$), Alice 2 ($\mathcal{A}_2$), Bob ($\mathcal{B}$), and Charlie ($\mathcal{C}$), who each perform one intervention in the course of each run. The experiment is set up in such a way that Charlie's intervention always occurs in the future lightcone of those of Alice 1 and 2, and Bob's intervention is spacelike-separated from those of the other agents (see Fig. 1a). Consider the following causal assumptions.

**Definite Causal Order.** There is a variable $\lambda$, taking a value on each run of the experiment, and associated partial orders $\prec_\lambda$ on $\{\mathcal{A}_1, \mathcal{A}_2, \mathcal{B}, \mathcal{C}\}$, such that on each run, the four agents are causally ordered according to $\prec_\lambda$ (cf. Ref. 3). (This means causal influences can propagate from agent $\mathcal{X}$ to $\mathcal{Y}$ only if $\mathcal{X} \prec_\lambda \mathcal{Y}$; here, causal influence is understood as an a priori notion, not directly linked to spacetime or correlations between variables.)

**Relativistic Causality.** The causal orders $\prec_\lambda$ respect the lightcone structure of the experiment. Concretely, $\mathcal{B}$ is $\prec_\lambda$-unrelated to $\mathcal{A}_1$, $\mathcal{A}_2$, and $\mathcal{C}$ for each $\lambda$ (because Bob acts at spacelike separation from the other agents; this rules out superluminal causation) and $\mathcal{C} \not\prec_\lambda \mathcal{A}_1$ and

$\mathcal{C} \not\prec_\lambda \mathcal{A}_2$ for each $\lambda$ (because Charlie acts in the future lightcone of Alice 1 and 2; this rules out retrocausation).

Without loss of generality, we will assume that $\lambda$ takes values in $\{1, 2\}$, where $\mathcal{A}_1 \prec_1 \mathcal{A}_2 \prec_1 \mathcal{C}$ and $\mathcal{A}_2 \prec_2 \mathcal{A}_1 \prec_2 \mathcal{C}$ (see Fig. 1b). (Strictly speaking, Relativistic Causality leaves open the possibility for other causal orders; their contribution to the argument is however already covered by $\prec_1$ and $\prec_2$. See Supplementary Note 2 for a proof of this fact and for more formal statements of the assumptions.) We now consider device-independent data in the form of correlations between classical settings $x_1, x_2, y, z$ and outcomes $a_1, a_2, b, c$ of the agents' interventions. The following third assumption imposes constraints on these correlations on the basis of the purely causal assumptions above.

**Free Interventions.** The settings $x_1, x_2, y, z$ have no relevant causes. In particular, they are (i) statistically independent of the hidden variable $\lambda$, and (ii) conditioned on any value of $\lambda$, statistically independent of any outcome variables of agents outside their $\prec_\lambda$-future. This means that agents cannot signal outside their $\prec_\lambda$-future, even when the value of $\lambda$ is known.

Part (i) of this assumption implies that the observed correlations, represented by a conditional probability distribution $p(a_1 a_2 bc | x_1 x_2 yz) =: p(\vec{a}bc | \vec{x}yz)$, can be written as

$$p(\vec{a}bc \mid \vec{x}yz) = \sum_{\lambda \in \{1,2\}} p(\lambda) p(\vec{a}bc \mid \vec{x}yz\lambda). \tag{1}$$

The no-signalling conditions of part (ii) can then be expressed as $p(\cdot \mid \cdot \lambda) \in \mathcal{DRF}_\lambda$, where

$$\mathcal{NS} := \{q \in \mathcal{P}_{\vec{a}bc|\vec{x}yz} : \vec{a}c \perp\!\!\!\perp_q y \text{ and } b \perp\!\!\!\perp_q \vec{x}z\}; \tag{2}$$

$$\mathcal{DRF}_1 := \{q \in \mathcal{NS} : a_1 b \perp\!\!\!\perp_q x_2 \text{ and } \vec{a}b \perp\!\!\!\perp_q z\}; \tag{3}$$

$$\mathcal{DRF}_2 := \{q \in \mathcal{NS} : a_2 b \perp\!\!\!\perp_q x_1 \text{ and } \vec{a}b \perp\!\!\!\perp_q z\}. \tag{4}$$

Here $\mathcal{P}_{\vec{a}bc|\vec{x}yz}$ is the set of conditional probability distributions, while $\perp\!\!\!\perp_q$ denotes statistical independence: for example, $\vec{a}c \perp\!\!\!\perp_q y$ means $\forall \vec{a}, c, y, y', z : \sum_b q(\vec{a}bc | \vec{x}yz) = \sum_b q(\vec{a}bc | \vec{x}y'z)$. $\mathcal{NS}$ is the set of correlations with no signalling between Bob and the other agents.

We will denote by $\mathcal{DRF}$ the set of all correlations $p(\vec{a}bc | \vec{x}yz)$ arising in experiments satisfying Definite Causal Order, Relativistic Causality, and Free Interventions—i.e. those of the form (1) with $p(\cdot | \cdot \lambda) \in \mathcal{DRF}_\lambda$. It is a polytope (see Methods), and is given by the convex hull

$$\mathcal{DRF} := \text{conv}(\mathcal{DRF}_1 \cup \mathcal{DRF}_2) \tag{5}$$

(see Fig. 1c).

A few comments about our three assumptions are in order. First of all, note that if a delay between the generation of the setting $x_1$ and outcome $a_1$ of Alice 1 is present, and two-way communication with Alice 2 during this period is allowed (or vice versa), then arbitrarily strong two-way signalling correlations between Alice 1 and 2 can arise. This includes correlations not in $\mathcal{DRF}$. Indeed, the Definite Causal Order assumption becomes interesting only when the agents' laboratories are assumed 'closed', in the sense that communication during such a delay (if present) is not allowed[3]. We do not formalise this here, but leave it open for discussion what violation of the inequalities derived below means in any given context.

Moreover, note that our notion of Relativistic Causality is relatively weak. Along with Free Interventions, it leads, for example, to the conditional distributions $p(\cdot | \cdot \lambda)$ being members of the $\mathcal{NS}$ polytope of Equation (2). This entails what is known as parameter

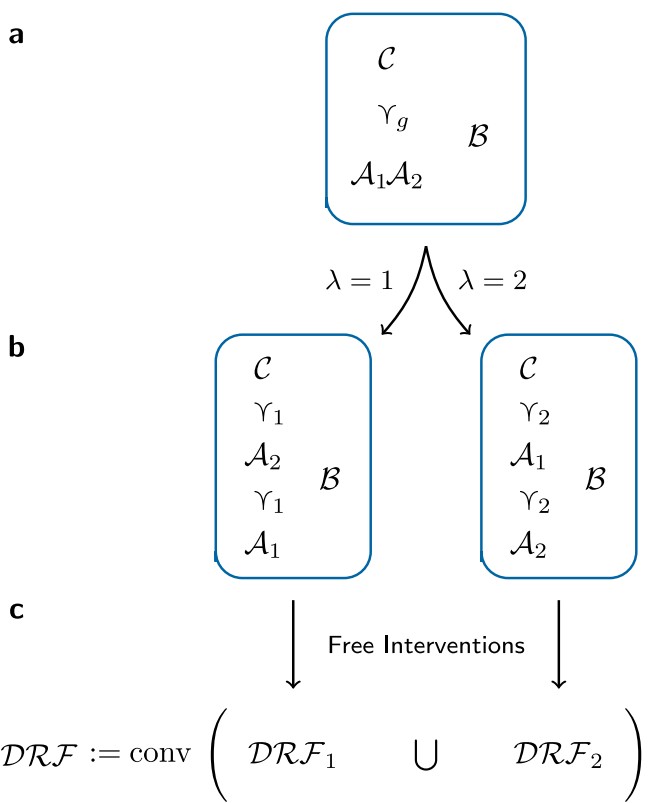

**a**

**b**

**c**

**Fig. 1 | Causal orders giving rise to the $\mathcal{DRF}$ polytope. a** An experiment is performed by Alice 1 ($\mathcal{A}_1$), Alice 2 ($\mathcal{A}_2$), Bob ($\mathcal{B}$), and Charlie ($\mathcal{C}$) in the spatio-temporal structure $\prec_g$ illustrated here: that is, Charlie always acts in the future lightcone of Alice 1 and Alice 2, and Bob acts at spacelike separation from the other agents. **b** The assumptions of Definite Causal Order and Relativistic Causality assert the existence of a variable $\lambda$ specifying a partial order $\prec_\lambda$ on all agents, such that $\prec_\lambda$ respects the spatiotemporal structure of **a**. (Other possibilities for $\prec_\lambda$, in which some of $\mathcal{A}_1$, $\mathcal{A}_2$ and $\mathcal{C}$ are unrelated, are not illustrated here as their contributions to $\mathcal{DRF}$ are already covered by $\prec_1$ and $\prec_2$.) **c** Conditioned on each value of $\lambda$, the Free Interventions assumption imposes statistical independence conditions, captured by the sets $\mathcal{DRF}_\lambda$, that rule out signalling outside the $\prec_\lambda$-future. $\mathcal{DRF}$ is the convex hull of $\mathcal{DRF}_1 \cup \mathcal{DRF}_2$, i.e. consists of the probabilistic mixtures of correlations in $\mathcal{DRF}_1$ and $\mathcal{DRF}_2$.

independence in the context of Bell's theorem[47]. It does, however, not entail Bell's stronger notion of Local Causality or Bell Locality, which is given by the conjunction of parameter independence and another condition known as outcome independence, and which (under Free Interventions) leads to Bell inequalities. (This is important, as it is already known that quantum correlations can violate Bell inequalities without any quantum switches being involved[45]).

Finally, in general one should allow for dynamical causal order, wherein the causal order between agents depends on interventions performed by agents in their causal past[6,37,48]. This would contradict part (i) of Free Interventions; however, since by Relativistic Causality no agents are in the causal past of Alice 1 and Alice 2, in our case this does not lead to any more general correlations than those already in the polytope $\mathcal{DRF}$ defined above. This is proved in Supplementary Note 2.

From now on, let us consider all variables $a_1, a_2, b, c, x_1, x_2, y, z$ to take values in $\{0, 1\}$. $\oplus$ denotes addition modulo 2. Moreover, to condense notation, we assume that the settings $x_1, x_2, y, z$ are independent and uniformly distributed (see Equation (9) in Methods for an example). The following inequality, together with its violation by the quantum switch demonstrated in the next section, forms our main result.

**Theorem 1.** If $p \in \mathcal{DRF}$ then

$$p(b=0, a_2 = x_1 \mid y=0) + p(b=1, a_1=x_2 \mid y=0) + p(b \oplus c = yz \mid x_1 = x_2 = 0) \leq \frac{7}{4},$$

(6)

and $\mathcal{DRF}$ saturates this bound.

**Proof sketch.** In an ordinary Bell scenario, any hidden variable model which is deterministic and satisfies conditions known as parameter independence (PI) and measurement independence (MI, also known as free choice) produces correlations that satisfy Bell inequalities (as determinism implies outcome independence). In a bipartite scenario with binary settings and outcomes, this can be strengthened: any hidden variable model satisfying PI and MI and in which just one of the measurement outcomes of one of the parties is predetermined by the hidden variable satisfies Clauser-Horne-Shimony-Holt (CHSH) inequalities. (This can be seen as a consequence of the monogamy of Bell nonlocality[49].) Inequality (6) is constructed in such a way that if the first two terms sum to unity and a hidden causal order $\lambda$ exists, then $\lambda$ must be perfectly correlated to the value of $b$ for the setting $y = 0$. Therefore, assuming PI and MI−which in our scenario follow from Relativistic Causality and Free Interventions−, Bob and Charlie cannot violate a CHSH inequality, bounding the third term by 3/4. More generally, for any $p \in \mathcal{DRF}$, a large value of the first two terms imposes a low upper bound on the value of the third term. This is made formal in Methods by way of a monogamy inequality.

An example of a deterministic correlation $p \in \mathcal{DRF}$ saturating (6) is defined by $a_1 = 0$, $a_2 = x_1$, $c = 0$ and $b = 0$; a nondeterministic example is given by setting $a_1 = 0$, $a_2 = x_1$ and letting Bob and Charlie use a PR box[50]. (PR correlations−which are maximally Bell-nonlocal yet non-signalling−are allowed in $\mathcal{DRF}$, as we do not assume full Bell Locality).

**Violation by the quantum switch**

The quantum switch is one of the few causally nonseparable processes that has a known physical interpretation, and the only such process to date that has been studied experimentally[29–34]. Yet, the device-independent correlations that it generates do not violate any causal inequalities as previously considered in literature[5,6]. (This is explained in more detail in Supplementary Note 1.) Here we will show that it does violate the inequality in Theorem 1.

The quantum switch can be described as a bipartite supermap[51], i.e. a map $\mathcal{S}$ taking two quantum operations $\mathcal{E}$, $\mathcal{F}$ on a system $T$, here taken

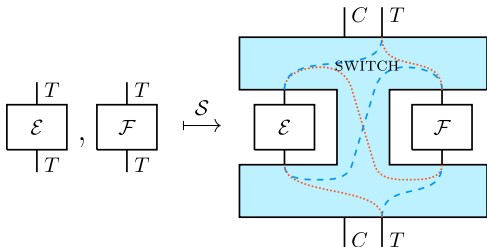

**Fig. 2 | The quantum switch.** Drawn here in blue, it is a bipartite supermap taking two quantum operations on the system $T$, denoted by $\mathcal{E}$ and $\mathcal{F}$, to an operation on $CT$, where $C$ is the control qubit (see Equation (7)). The dotted (red) and dashed (blue) lines illustrate the wirings to which the quantum switch reduces upon preparation of $C$ in state $|0\rangle\langle0|$ and $|1\rangle\langle1|$, respectively.

to be a qubit, to an operation $\mathcal{S}(\mathcal{E}, \mathcal{F})$ on the joint system $CT$, which applies $\mathcal{E}$ and $\mathcal{F}$ to the target system $T$ in an order that is coherently controlled by the state of a control qubit $C$ (see Fig. 2). Hence, if these systems are described by Hilbert spaces $\mathcal{H}_T \cong \mathcal{H}_C \cong \mathbb{C}^2$ and if $\mathcal{E}(\cdot) = E(\cdot)E^\dagger$ and $\mathcal{F}(\cdot) = F(\cdot)F^\dagger$ are pure operations described by the Kraus operators $E, F : \mathcal{H}_T \to \mathcal{H}_T$, then $\mathcal{S}(\mathcal{E}, \mathcal{F})(\cdot) = W(\cdot)W^\dagger$ where $W : \mathcal{H}_C \otimes \mathcal{H}_T \to \mathcal{H}_C \otimes \mathcal{H}_T$ is the operator defined by[2]

$$W := |0\rangle\langle0| \otimes FE + |1\rangle\langle1| \otimes EF.$$

(7)

To see how the four agents discussed in the previous section can violate Inequality (6) when they have access to a quantum switch, we prepare the target system in the initial state $|0\rangle_T$ while entangling the input control qubit $C$ to an additional qubit $B$ in the state $|\Phi^+\rangle := (|00\rangle + |11\rangle)/\sqrt{2}$ (see Fig. 3). Alice 1 and Alice 2, placed inside the two slots of the switch, use measure-and-prepare instruments: for $i = 1, 2$, Alice $i$ measures the incoming target system $T$ in the computational basis −independently of her setting $x_i$−and records the outcome in $a_i$. She then prepares $T$ in the computational basis state $|x_i\rangle$, before sending it away. Bob has access to the spacelike-separated qubit $B$, which he measures in the computational ($Z$) direction if $y = 0$, and in the $X$ direction if $y = 1$; he records his outcome in $b$. Finally, Charlie measures the output control qubit $C$ in the $Z + X$ (for $z = 0$) or $Z − X$ (for $z = 1$) direction, recording his outcome in $c$. The output target system is discarded.

With these choices of instruments and state preparations, the first two terms in Inequality (6) are both 1/2. For instance, if $y = 0$, Bob obtains $b = 0$ with probability 1/2; and postselecting on that outcome yields the same correlations in the switch as if the control qubit had been prepared in state $|0\rangle_C$. The latter would reduce the switch to a wiring in which Alice 1 is before Alice 2, meaning that $a_2 = x_1$. (Similarly for the second term.) For the third term of (6), note that if $x_1 = x_2 = 0$ then Alice 1 and 2 will both measure and reprepare the target system to be in state $|0\rangle_T$; in particular, their operations commute on the initial target state $|0\rangle_T$, so that the state of the control system is unaffected. This means that Bob and Charlie perform an ordinary Bell test on the maximally entangled state $|\Phi^+\rangle_{CB}$ (see Eq. (17) in Methods). With the choice of measurement directions given above, this yields a CHSH value of $1/2 + \sqrt{2}/4$, so that Inequality (6) is violated:

$$\frac{1}{2} + \frac{1}{2} + \left(\frac{1}{2} + \frac{\sqrt{2}}{4}\right) \approx 1.8536 > \frac{7}{4}.$$

(8)

This shows that the correlations observed in this quantum switch setup do not admit a hidden variable model satisfying Equations (1)–(4), thus establishing indefinite causal order in the quantum switch under the assumptions of Relativistic Causality and Free Interventions. Equation (8) is in fact the maximal quantum violation of Inequality (6) in this scenario, or indeed in any quantum scenario where Bob's observables commute with Alice's and Charlie's: this follows from the

Tsirelson bound[52] and the fact that the algebraic maximum of the first two terms is 1.

**More DRF inequalities**

Table 1 presents some more inequalities that are valid and tight for the polytope $\mathcal{DRF}$ of correlations admitting a hidden variable model satisfying Definite Causal Order, Relativistic Causality, and Free Interventions. The inequalities listed here do not involve Charlie's measurement setting $z$; thus, they define faces (though not necessarily

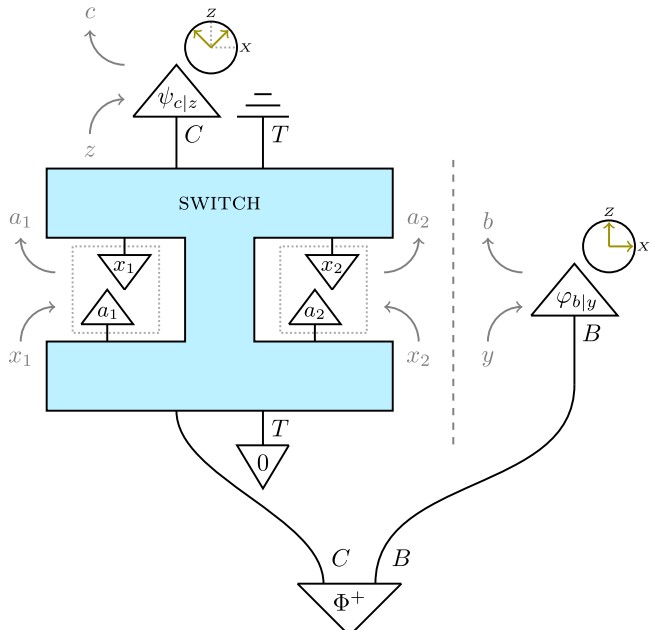

**Fig. 3 | The quantum switch setup violating Inequality (6).** The input-output direction in this diagram is from bottom to top. The switch's control system $C$ and a system $B$ held by Bob are prepared in the maximally entangled state $|\Phi^+\rangle$. The target system $T$ is prepared in state $|0\rangle$, measured and reprepared in the computational basis by Alice 1 and 2 (in the dotted boxes), and ultimately discarded. Finally, Bob and Charlie perform, for each of their settings $y$ and $z$, projective measurements on $B$ and the output control system $C$ in directions in the $XZ$ plane of the Bloch sphere indicated by the green arrows. $\langle\varphi_{b|y}|$ is the effect corresponding to Bob observing outcome $b$ upon setting $y$; similarly for Charlie's effect $\langle\psi_{c|z}|$. The diagram as a whole defines the probability $p(\bar{a}bc\,|\,\bar{x}yz)$, also given in Equation (16) in Methods.

facets) of a lower-dimensional version of $\mathcal{DRF}$, which is more amenable to computational analysis and to experimental tests of inequality violations. The polytope and the faces listed here are discussed in more detail in Methods.

Inequalities (i)–(iii) are similar to Inequality (6), and are (weakly) violated by the quantum switch using the same setup as described earlier and depicted in Fig. 3, but with $z$ fixed to 0. To understand inequality (iv), observe that the Alices can use their measure-and-prepare instruments to effectively perform a computational basis measurement of the input control qubit, with outcome $a_1$, by setting $x_2 = 1$. Indeed, in this case, Alice 2 prepares the target system in state $|1\rangle_T$, while it was initially prepared in state $|0\rangle_T$; therefore, each value of $a_1$ is only compatible with one of the computational basis states of the control qubit $C$ (see Eq. (18) in Methods). This observation suggests that the argument in the Proof Sketch of Theorem 1 can also be applied to correlations between the causal order variable $\lambda$ and the outcome $a_1$, rather than $b$. This is witnessed by Inequality (iv). Its first three terms are constructed in such a way that a high value for them implies a strong correlation between $\lambda$ and $a_1$ for the settings $x_1 = x_2 = 1$, thereby bounding the final CHSH term, which now involves $a_1$. In the quantum switch, on the other hand, Alice's measure-and-prepare instruments yield the maximum value of 1 for the first three terms, while their effective $Z$ measurement of the control qubit described above contributes to a high value for the CHSH term, thus violating Inequality (iv). With appropriate measurement directions for Bob ($Z + X$ and $Z - X$) and Charlie ($X$), it is violated up to the quantum bound, just like Inequality (6). Merits of Inequality (iv) as compared to (6) are however that it does not involve a setting for Charlie and that its proof relies on mathematically weaker assumptions (see Methods).

The final four inequalities in Table 1 show the similarity between the facets of the bipartite causal polytope studied in previous literature[36] ((v) and (vii)) and some of the facets of $\mathcal{DRF}$ ((vi) and (viii)), thus highlighting one consequence of adding the Relativistic Causality assumption. None of these inequalities can however be violated by the quantum switch, because they do not involve the variable $c$ (see Supplementary Note 1). They are discussed in more detail in Methods.

## Discussion

The quantum switch, when considered in isolation, does not violate causal inequalities as previously defined in the literature[5,6]. As a result, it has long been believed that the indefinite causal order of the quantum switch does not in general admit a device-independent certification. The present result, however, shows that such a certification is possible when the set of allowed causal orders is constrained. In our case these

**Table 1 | Some inequalities following from Definite Causal Order, Relativistic Causality, and Free Interventions**

| | Face-defining inequality | Dimension |
|---|---|---|
| (i) | $p(b = 0, a_2 = x_1 \mid y = 0) + p(b = 1, a_1 = x_2 \mid y = 0) + p(b \oplus c = x_2 y \mid x_1 = 0) \le 7/4$ | 67 |
| (ii) | $p(b = 0, a_2 = x_1 \mid \mathbf{x_2}y = \mathbf{0}0) + p(b = 1, a_1 = x_2 \mid \mathbf{x_1}y = \mathbf{0}0) + p(b \oplus c = x_2 y \mid x_1 = 0) \le 7/4$ | 73 |
| (iii) | $p(b = 0, a_2 = x_1 \mid x_2 y = 00) + p(b = 1, a_1 = x_2 \mid \mathbf{x_1}y = \mathbf{1}0) + p(b \oplus c = x_2 y \mid x_1 = 0)$ $+ \mathbf{p(a_2 = 1, c \oplus 1 = b = y \mid x_1 x_2 = 00)} \le 7/4$ | **85** |
| (iv) | $1/2\left[p(a_1 = 0 \mid x_1 x_2 = 10) + p(a_2 = 0 \mid x_1 x_2 = 01) - p(a_1 a_2 = 00 \mid x_1 x_2 = 11)\right]$ $+ p((x_2 a_1 + (x_2 \oplus 1)c) \oplus b = x_2 y \mid x_1 = x_2) \le 7/4$ | 61 |
| (v) | $p(a_1 = x_2, a_2 = x_1) \le 1/2$ | 83 |
| (vi) | $p(a_1 = x_2, a_2 = x_1, \mathbf{b = 0} \mid \mathbf{y = 0}) + \mathbf{1/2}\,p(b = 1 \mid y = 0) \le 1/2$ | **85** |
| (vii) | $p(x_1(a_1 \oplus x_2) = 0, x_2(a_2 \oplus x_1) = 0) \le 3/4$ | 83 |
| (viii) | $p(x_1(a_1 \oplus x_2) = 0, x_2(a_2 \oplus x_1) = 0, \mathbf{b = 0} \mid \mathbf{y = 0}) + \mathbf{3/4}\,p(b = 1 \mid y = 0) \le 3/4$ | **85** |

These are inequalities valid and tight for $\mathcal{DRF}$—an 86-dimensional version of the polytope without Charlie's setting $z$, defined in Equation (19) in Methods—violation of which thus indicates falsification of the conjunction of Definite Causal Order, Relativistic Causality, and Free Interventions. Inequalities (i)–(iv) are violated by the quantum switch, whereas (v)–(viii) are satisfied by all quantum switch correlations. The inequalities are listed along with the dimensions of the faces of they support; each 85-dimensional face constitutes a facet of the polytope. The boldface highlights aspects in which an inequality differs from the preceding one. For conciseness, we assume that all settings are independently and uniformly distributed (see Equation (9)). Inequalities (iii), (vi), and (viii) were found computationally, while the others were derived analytically.

constraints arise from spatiotemporal information together with a Relativistic Causality assumption ruling out influences outside the future lightcone, but the constraints could be motivated differently too (e.g. by the topology of an experimental setup). Together with Free Interventions, these causal constraints impose conditions akin to what is known as parameter independence in the context of Bell's theorem.

We arrived at this result by deriving an inequality and exhibiting a quantum switch setup violating it up to the quantum bound. The intuition behind this violation lies in the fact that in our setup, one of Bob's outcomes is simultaneously correlated to the causal order in the switch (if such a causal order is assumed to exist) and to Charlie's measurements in such a way that Bob and Charlie violate a CHSH inequality. The monogamy of Bell nonlocality tells us that such simultaneous correlations can only arise when one of Relativistic Causality and Free Interventions is violated.

Note that violation of our inequalities requires violation of a CHSH inequality. As such, they cannot be violated by classical processes subject to the same spatiotemporal constraints. This sets them apart from regular causal inequalities, which can be violated by both quantum and classical processes[38] and can therefore not distinguish between classical and nonclassical indefinite causality.

It is worth noting that locality assumptions like Relativistic Causality have already been used in discussions of indefinite causal order. In Ref. [6], for example, a causal correlation is a convex sum of correlations compatible with a (possibly dynamical) configuration of parties in spacetime, where each term involving spacelike separation is assumed to involve no superluminal signalling. Relative to this, the novelty of the present work lies in making use of available partial information about causal and spatiotemporal relations—viz. that Charlie is after the Alices and Bob is spacelike to the Alices and Charlie—rather than allowing arbitrary causal orders. Another locality notion has been studied in the context of Bell's theorem for temporal order[27,33]. Here, violation of a Bell inequality is argued to imply indefinite causal order for the quantum switch under suitable separability and locality assumptions. This method is, however, not device-independent, as these assumptions rely on descriptions of states and transformations rather than just the observed correlations.

It is natural to wonder about the consequences of experimental violation of the inequalities derived here. Most current implementations of the quantum switch are based on optical interferometric setups sending a photon along a superposition of paths passing through Alice 1 and 2's devices in different orders[29–33]. When used to probe correlations of measurement outcomes, these setups require the outcomes $a_1, a_2$ to be read out only at the end, i.e. after both photon paths have passed through both Alices' devices, in order not to destroy the superposition of causal orders[30,32]. These delayed measurements, however, mean that both outcomes only obtain a definite value in the intersection of the future lightcones of the spacetime loci where $x_1$ and $x_2$ are chosen. Therefore, violation of one of our inequalities by such experiments would, at least from the point of view of classical relativity theory, not demonstrate an interesting notion of indefinite causal order: it comes as no surprise that $x_1$ can influence $a_2$ while at the same time $x_2$ influences $a_1$. This ties into the broader debate of whether these photonic experiments realise the quantum switch or merely simulate it[33,42,53–56]. We note that considerations involving gravitational quantum switch implementations and/or quantum reference frames[27,55,57] may offer different perspectives on this problem.

Provided one succeeds in avoiding this and other loopholes, experimental violation of the inequalities derived here could put restrictions on possible theories of quantum gravity compatible with observation. On a more practical level, an interesting direction of future research is to determine whether these inequalities could be used for new device-independent protocols, analogously to how Bell's theorem is used for device-independent quantum key distribution[58].

The technique by which we utilise the Relativistic Causality assumption and Bell inequalities for our certification was inspired by recent results on Wigner's friend scenarios[59], and might be applicable to the certification of other phenomena as well. It also suggests follow-ups on this work, such as proving the violation of our three assumptions without inequalities (cf. the GHZ test[60] or Hardy's test[61]) or without settings (cf. Bell nonlocality in networks[62]). Finally, a natural extension of our result is to demonstrate the violation of appropriately generalised inequalities by processes beyond the quantum switch. For instance, it is known that any pure entangled state violates a Bell inequality[63]. Could it likewise be true that all unitary[7,40] causally non-separable quantum processes violate a device-independent inequality witnessing their causal indefiniteness?

## Methods
### Proof of Theorem 1
We assume that all settings are binary and uniformly and independently distributed. This allows us to use shorthands such as

$$p(b=0, a_2 = x_1 \mid y=0) := \frac{1}{8} \sum_{x_1,x_2,z \in \{0,1\}} p(b=0, a_2=x_1 \mid x_1 x_2 z, y=0) \tag{9}$$

and

$$p(b \oplus c = yz) := \frac{1}{4} \sum_{y,z \in \{0,1\}} p(b \oplus c = yz \mid yz). \tag{10}$$

This assumption is made purely to simplify notation; it is not a physical requirement and plays no role in the proof below.

**Proof of Theorem 1.** Recall the definitions of $\mathcal{DRF}_1$ and $\mathcal{DRF}_2$ in Equations (3) and (4). Because $\mathcal{DRF}$ is the convex hull of $\mathcal{DRF}_1$ and $\mathcal{DRF}_2$ and Inequality (6) is linear, it suffices to prove the inequality for the latter two polytopes individually. We give the proof for $\mathcal{DRF}_1$; the case for $\mathcal{DRF}_2$ is analogous.

Suppose $p \in \mathcal{DRF}_1$, and denote the first two terms of the inequality by $\alpha$:

$$\alpha := p(b=0, a_2 = x_1 \mid y=0) + p(b=1, a_1 = x_2 \mid y=0). \tag{11}$$

Note that

$$p(b=0, a_2 = x_1 \mid y=0) \le p(b=0 \mid y=0) \tag{12}$$

and, because $a_1 b \perp\!\!\!\perp_p x_2$ for $p \in \mathcal{DRF}_1$,

$$p(b=1, a_1 = x_2 \mid y=0) = \frac{1}{2} p(b=1 \mid y=0). \tag{13}$$

Adding Equations (12) and (13) and rewriting gives

$$p(b=0 \mid y=0) \ge 2\alpha - 1. \tag{14}$$

The monogamy of Bell nonlocality, however, tells us that for nonsignalling correlations, a highly probable outcome is incompatible with a large CHSH value. More precisely, applying the monogamy inequality of Ref. [49] to the correlation $p(bc \mid yz, x_1 = x_2 = 0)$ (and noting that $b \perp\!\!\!\perp_p x_1 x_2 z$) shows that the last term of Inequality (6) is bounded as

$$\begin{aligned}
p(b \oplus c = yz \mid x_1 = x_2 = 0) &\le \frac{5}{4} - \frac{1}{2} p(b=0 \mid y=0) \\
&\le \frac{5}{4} - \frac{1}{2}(2\alpha - 1) = \frac{7}{4} - \alpha,
\end{aligned} \tag{15}$$

where we used Equation (14) for the second inequality. Combining this with Equation (11) completes the proof.

(It is worth noting that the restriction that $a_1a_2b \perp\!\!\!\perp_p z$ in $\mathcal{DRF}_{1,2}$, corresponding to the assumption that Charlie is in the causal future of Alice 1 and 2, is not used in the proof of Theorem 1. However, including it yields a polytope that more accurately reflects the set of correlations that can arise in the scenario under consideration. Note also that it is essential that Charlie is not in the causal past of Alice 1 or 2, for this excludes the possibility that the causal order between Alice 1 and 2 depends on $z$.)

### The quantum switch correlations

Here we analyse in more detail the correlations generated by the quantum switch in the scenario depicted in Fig. 3, making more rigorous our claims that Charlie and Alice 1 can effectively measure the input control system $C$.

The interventions that we consider have single Kraus operators for each classical outcome: Alice $i$'s Kraus operator corresponding to measuring $a_i$ and preparing $x_i$ is given by the linear operator $|x_i\rangle\langle a_i| : \mathcal{H}_T \to \mathcal{H}_T$, while Bob's and Charlie's projective measurements are described by the effects $\langle\varphi_{b|y}| : \mathcal{H}_B \to \mathbb{C}$ and $\langle\psi_{c|z}| : \mathcal{H}_C \to \mathbb{C}$, respectively, whose directions in the Bloch sphere are indicated in Fig. 3. The setting-outcome correlation corresponding to the scenario depicted in Fig. 3 is then given by Equation (7) and the Born rule:

$$p(\vec{a}bc \mid \vec{x}yz) = \left\| \langle\psi_{c|z}|_C \langle\varphi_{b|y}|_B (|0\rangle\langle0|_C \otimes |x_2\rangle\langle a_2|x_1\rangle\langle a_1|_T \right.$$
$$\left. + |1\rangle\langle1|_C \otimes |x_1\rangle\langle a_1|x_2\rangle\langle a_2|_T)|\Phi^+\rangle_{CB}|0\rangle_T \right\|^2. \tag{16}$$

Note first of all that if $x_1 = x_2 = 0$, this reduces to

$$p(\vec{a}bc \mid 00yz) = \left|\left(\langle\psi_{c|z}|_C \langle\varphi_{b|y}|_B\right)|\Phi^+\rangle_{CB}\right|^2 \delta_{a_1 = a_2 = 0}; \tag{17}$$

thus, Bob and Charlie effectively perform a normal Bell test on $|\Phi^+\rangle_{CB}$, yielding the maximum quantum value of $1/2 + \sqrt{2}/4$ for the third term in Inequality (6), thereby violating it.

On the other hand, if $x_2 = 1$ then the marginal distribution over $a_1$ and $b$ reduces to

$$p(a_1 b \mid x_1, x_2 = 1, y) = \left|\left(\langle a_1|_C \langle\varphi_{b|y}|_B\right)|\Phi^+\rangle_{CB}\right|^2, \tag{18}$$

showing that Alice 1's measurement yields the same correlations as a computational basis measurement of $C$, as we claimed in our discussion of Inequality (iv) in Table 1.

### Vertices of $\mathcal{DRF}$

A polytope $\mathcal{X} \subseteq \mathbb{R}^d$ is a convex body with flat sides; it can be described either as the convex hull of a finite set of points, or as the intersection of finitely many closed halfspaces—i.e. the set of points satisfying a finite collection of linear inequalities—as long as this intersection is bounded[64]. The vertices of $\mathcal{X}$ are its extremal points. We call a linear inequality $\alpha^T x \le \beta$, for $\alpha \in \mathbb{R}^d$ and $\beta \in \mathbb{R}$, valid for $\mathcal{X}$ if it holds for all $x \in \mathcal{X}$, and tight if equality holds for some $x \in \mathcal{X}$. Each linear inequality defines a hyperplane $\{x \in \mathbb{R}^d : \alpha^T x = \beta\}$; if the inequality is valid for $\mathcal{X}$, the intersection of this hyperplane with $\mathcal{X}$ is a face of $\mathcal{X}$, which is itself a polytope. If the dimension of a face is one less than the dimension of $\mathcal{X}$ itself, we call the face a facet. Any polytope is completely determined by the set of all its facets, or equivalently, its facet-defining inequalities.

We focus on the variant of $\mathcal{DRF}$ with binary settings and outcomes and without Charlie's setting $z$, defined by

$$\mathcal{NS} := \{p \in \mathcal{P}_{\vec{a}bc|\vec{x}y} : \vec{a}c \perp\!\!\!\perp_p y \text{ and } b \perp\!\!\!\perp_p \vec{x}\}; \tag{19}$$

$$\mathcal{DRF}_1 := \{p \in \mathcal{NS} : a_1 b \perp\!\!\!\perp_p x_2\}; \tag{20}$$

$$\mathcal{DRF}_2 := \{p \in \mathcal{NS} : a_2 b \perp\!\!\!\perp_p x_1\}; \tag{21}$$

$$\mathcal{DRF} := \text{conv}(\mathcal{DRF}_1 \cup \mathcal{DRF}_2). \tag{22}$$

Here conv denotes the convex hull. Note that $\mathcal{DRF}_{1,2} \subsetneq \mathcal{DRF} \subsetneq \mathcal{NS} \subsetneq \mathcal{P}_{\vec{a}bc|\vec{x}y}$, and that these are polytopes: all except $\mathcal{DRF}$ are defined uniquely by linear no-signalling and normalisation constraints and non-negativity of probabilities.

$\mathcal{DRF}_1$ ($\mathcal{DRF}_2$) is 80-dimensional and admits a facet description in terms of 128 facets corresponding to non-negativity of probabilities. Using the software PANDA[65], we converted this facet description into a vertex description, exploiting symmetries of the polytope for efficiency. Taking the vertices of $\mathcal{DRF}_1$ and $\mathcal{DRF}_2$ together then yields the 9165312 vertices of $\mathcal{DRF}$, which fall into 219 equivalence classes under symmetries of $\mathcal{DRF}$. These symmetries correspond to interchanging Alice 1 and 2 and to relabelling the seven binary variables, possibly depending on the values of preceding variables in the causal order. More precisely, a minimal generating set of the symmetry group we used is induced by the following relabellings:

$$\begin{aligned} x_1 &\mapsto x_1 \oplus 1, \quad a_1 \mapsto a_1 \oplus x_1, \\ x_2 &\mapsto x_2 \oplus 1, \quad a_2 \mapsto a_2 \oplus x_2, \\ c &\mapsto c \oplus a_1 a_2 x_1 x_2, \\ y &\mapsto y \oplus 1, \quad b \mapsto b \oplus y, \\ (a_1, a_2, x_1, x_2) &\mapsto (a_2, a_1, x_2, x_1). \end{aligned} \tag{23}$$

Only 3 of the vertex classes of $\mathcal{DRF}$ are deterministic and therefore local; the others are nonlocal and have probabilities that are multiples of 1/2. The vertices also tell us that $\mathcal{DRF}$ is 86-dimensional, matching the dimension of the ambient no-signalling polytope $\mathcal{NS}$.

### Inequalities in Table 1

We will now discuss the inequalities in Table 1 in a bit more detail. These inequalities are valid and tight for and thus define faces of the 86-dimensional polytope defined in Equation (19).

Inequality (i) in Table 1 is similar to (6), except that $z$ is replaced by $x_2$ in the CHSH term. The proof that (i) is valid and tight for $\mathcal{DRF}$ is directly analogous to the proof of Theorem 1. It is weakly violated by the quantum switch setup described in the main text, fixing $z = 0$, which yields a value of $1.7652 > 7/4$. A stronger violation can be found by using the observation, pointed out in the main text, that if $x_2 = 1$, then the probabilities for $a_1$ coincide with those of a computational basis measurement of the input control system. In particular, optimising over projective qubit measurements for Bob and Charlie, denoting Charlie's measurement outcome by $c'$, and letting Charlie output $c := x_2 a_1 + (x_2 \oplus 1)c'$ leads to a value of (i) of approximately 1.8274.

Inequality (ii) differs from (i) in the respect that the first two terms are conditioned on the values of $x_1$ and $x_2$. The violations by the quantum switch correlations discussed in this paper are unaffected by this change. What makes (ii) interesting is that it only depends on the probabilities of $a_i$ when $x_i = 0$, for $i = 1, 2$. Moreover, if we adopt the strategy for Charlie described in the previous paragraph, the outcome $c$ of Charlie's measurement is only needed when $x_1 = x_2 = 0$. This poses an experimental advantage, as it reduces the number of measurements to be made. Geometrically, it entails that there is a still lower-dimensional polytope, which can be violated by the quantum switch, namely where $a_i$ ($c$) only takes values when $x_i = 0$ ($x_1 = x_2 = 0$).

Although it is in principle possible to compute all facets of $\mathcal{DRF}$ from its known vertex description, in practice this is complicated by its high dimension and high number of vertices. However, the dimension of known faces, such as those defined by the inequalities in Table 1, can be determined by counting the number of affinely independent vertices saturating the inequality (and subtracting 1). Moreover, the knowledge of the vertices can be used to pivot high-dimensional faces onto adjacent facets. Inequality (*iii*) has been obtained by pivoting a variant of Inequality (*ii*) in this way. Its additional fourth term, however, vanishes for all quantum switch correlations discussed in this paper, thus not paving the way for stronger inequality violations.

Inequality (*iv*) is motivated in the main text and proved in Supplementary Note 3. The assumptions required for this proof are strictly weaker than those required for Inequalities (6), (*i*), and (*ii*): namely, while the latter inequalities require the joint independence $a_1 b \perp\!\!\!\perp_p x_2$ (see Equation (13)) to hold in $\mathcal{DRF}_1$, the proof of (*iv*) only requires $a_1 \perp\!\!\!\perp_p x_2$ and $b \perp\!\!\!\perp_p x_2$ separately (see Supplementary Equation (15)). Similarly for $\mathcal{DRF}_2$. This can be considered physically desirable because it separates the no-signalling constraints imposed by the Relativistic Causality condition from those imposed by the order between Alice 1 and 2 (which might involve exotic effects not in accordance with relativity theory).

The final four inequalities in Table 1 highlight the similarity between $\mathcal{DRF}$ and the bipartite causal polytope studied in e.g. Ref. 36. The latter consists of causal correlations $p(a_1 a_2|x_1 x_2)$, i.e. those that can be written as

$$p = \mu p^1 + (1 - \mu) p^2, \tag{24}$$

where $\mu \in [0,1], a_1 \perp\!\!\!\perp_{p^1} x_2$ and $a_2 \perp\!\!\!\perp_{p^2} x_1$. The causal inequality (*v*), referred to as a 'guess your neighbour's input' inequality, defines one of the two inequivalent nontrivial facets of the bipartite causal polytope. Note that by our Definite Causal Order assumption, any correlation $p \in \mathcal{DRF}$ has a causal marginal $p(a_1 a_2|x_1 x_2)$, so that (*v*) is also valid for $\mathcal{DRF}$. However, it is no facet of $\mathcal{DRF}$; instead, (*vi*) is a facet adjacent to the face defined by (*v*), obtained by pivoting (*v*) onto the vertices of $\mathcal{DRF}$ as described above. Inequality (*vi*) can be interpreted as a bound on the winning probability in a game where the parties get one point if Alice 1 and 2 correctly guess each other's input and Bob outputs 0, while they get half a point if Bob outputs 1. This makes the effect of Relativistic Causality apparent: without it, Bob's output could be influenced by the correctness of Alice's guesses, yielding a value of up to 3/4 for (*vi*) even when Alice's marginal distribution is causal, and thus does not violate (*v*). (As an example, consider the deterministic distribution $a_1 = 0$, $a_2 = x_1$, $b = x_2$, equally mixed with $a_1 = 1$, $a_2 = x_1$, $b = x_2 \oplus 1$ if one requires that the total distribution still be nonsignalling between spacelike-separated parties).

Note that any bipartite quantum process violating the causal inequality (*v*)[36] can be trivially extended to a four-partite quantum process violating (*vi*). On the other hand, neither of these inequalities can be violated by the quantum switch, as they are independent of $c$ and discarding the output control qubit renders the switch causally separable[2,5] (see Supplementary Note 1). Similar results hold for the other facet of the causal polytope: see (*vii*) and (*viii*).

**Notes added at proof.** After the completion of this work, the authors have become aware of partial overlap with results obtained in an independent work of Gogioso and Pinzani[48]. That work proposes a device-independent framework, which captures causal indefiniteness in partially definite causal orders. The authors use numerical methods to show that the quantum switch exhibits indefinite causal order in a scenario similar to the one presented in this article. Our work provides an analytical approach to certify this indefinite causal order via the violation of an inequality.

## Data availability
The sets of vertices of the $\mathcal{DRF}$ and $\mathcal{DRF}_1$ polytopes, computationally generated and analysed during this study, are available in Zenodo under accession code 7564812.

## Code availability
The code used to analyse the polytopes and quantum switch correlations is also available in Zenodo under accession code 7564812. This was used in conjunction with the publicly available software PANDA[65].

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

## Acknowledgements

T.vd.L. is grateful to Nikolai Miklin, Nick Ormrod, and Marc-Olivier Renou for insightful discussions. This work was supported by the Hong Kong Research Grant Council through the Senior Research Fellowship Scheme SRFS2021-7S02 and the Research Impact Fund R7035-21F, by the Croucher Foundation, and by the John Templeton Foundation through the ID# 62312 grant, as part of the 'The Quantum Information Structure of Spacetime' Project (QISS). The opinions expressed in this publication are those of the authors and do not necessarily reflect the views of the John Templeton Foundation. Research at the Perimeter Institute is supported by the Government of Canada through the Department of Innovation, Science and Economic Development Canada and by the Province of Ontario through the Ministry of Research, Innovation and Science. For the purpose of Open Access, the author has applied a CCBY public copyright licence to any Author Accepted Manuscript (AAM) version arising from this submission.

## Author contributions

T.vd.L., J.B. and G.C. contributed to the research leading to this work and prepared the manuscript. J.B. and G.C. supervised the project and contributed equally.

## Competing interests

The authors declare no competing interests.
