## [Peer Review File · Nature Communications]

Device-independent certification of indefinite causal order in the quantum switchREVIEWER COMMENTS

Reviewer #1 (Remarks to the Author):

The submitted manuscript considers a central and fundamental question concerning causality in quantum theory, “Is it possible to have Device-independent certification of indefinite causal order within quantum systems?”. This paper provides substantial progress by presenting a fair scenario where the quantum switch allows device-independent certification of indefinite causal order. Apart from an important detail discussed below, the paper is well written, clear, and direct. Additionally, the methods and ideas are presented in a very accessible manner. I expect this work to have a positive impact and I strongly recommend for publication (see however the comments below).

Technical comments/suggestions:

For this work, the authors have used the software PANDA to obtain all the facets of the LC polytope in this dichotomic given scenario. They have also organized the inequalities in 219 equivalence classes. However, they do not provide their code, nor their data. It would be useful and practical for the community to have a file with all the vertices and inequalities. Not providing their code for sorting the inequalities and data (vertices and inequalities of LC) go against the transparency of science and the open source philosophy.

The authors should provide the code/data used in this work in some online repository.

In the abstract, we can read:

“Despite previous evidence suggesting a negative answer”

and in the discussion section, we read:

“It has long been believed that the indefinite causal order in the quantum switch does not admit a device-independent certification [5, 26, 28, 32, 33, 37, 40, 42, 49]. In contrast to this belief, the present result shows that the conjunction of free randomness, definite causal order, and locality (in terms of parameter independence) is inconsistent with correlations observed in the presence of the quantum switch”

In this current form, the submitted manuscript suggests that previous works may have wrong claims, or that they were mere evidence that device-independent certification of indefinite causality with the quantum switch is not possible. However, this is not true. There are rigorous and proper mathematical proofs of this fact. The key point here is that the definition of device-independent certification used in previous works is not the same used in the submitted manuscript. The submitted manuscript also required an extra hypothesis, the hypothesis of locality. I understand that this is a fair hypothesis which is well justified by the submitted manuscript. Also, the expression “device-independence” may be used in a broader and rather vague manner. However, it is important to re-write some sentences of the submitted manuscript to avoid this misconception. It should be clear for the readers that the definition of device-independent used to prove previous theorems is not the same definition of device-independent used in the submitted manuscript.

I’d say that the main merit of this work is to identify how to make use of LC scenario.

With that in mind, Eqs.27,28,29,30 are very enlightening. They summarize the scenario in a clear and direct manner. I’d suggest the authors to include it in the main text. Also, I’d suggest the authors to add a picture to illustrate LC1 and LC2, this would make the manuscript more accessible for readers from a different area. Fig.3 is very enlightening to clarify the quantum switch strategy, but Fig.1 does not fully clarify the scenario.

Line 109: Although the term PR box is well known to the Bell NL community, the authors should define it or at to cite a reference here.

Line 125: It could be useful for the readers if the authors explicitly states that Alices’ measurement choice does not depend on their input x_i for instance, something like “Independently of the classical input x_i , Alice i measures the incoming target system T in the computational basis,”

Line 263: It is not clear what is the overlap of the submitted manuscript with Ref.61 (Ref.61 has 474 pages!). The authors could write a couple of sentences on that.

Reviewer #2 (Remarks to the Author):

The submitted manuscript considers an important problem of certifying indefinite causal order. When it comes to experimental certification scientists could hope to certify a process with indefinite causal order that can be actually implemented in a laboratory. If this is to be done in the black box scenario, the obvious choice would be the violation of causal inequality. However, the big problem for the community is that all indefinite causal order processes that are implementable, most notably the quantum switch, for now, cannot violate causal inequalities (akin to entangled states which could not violate any Bell inequalities). Some previous results exist in semi-device-independent certification of indefinite causal order.

Here, the authors for the first time show a way for device-independent certification of a process that has indefinite causal order but does not violate any causal inequality. They develop an inequality, satisfied by all processes with definite causal order, and show that quantum switch can be used to provide a violation. This is done by introducing two additional parties, one that is spacelike separated from the main parties undergoing quantum switch and another one in the future cone of the "main" parties. It is these two additional parties that violate CHSH inequality only if the main parties are subject to indefinite causal order.

I think that the paper is very interesting, the result is important and could be motivation for further research in a similar direction. I quite like the idea of definite causal order preventing CHSH violation and I think this submission merits publication in Nature Communications after clarifying a few things.

I have a few suggestions/questions:

- 1) The authors several times claim that there are fewer assumptions than in Bell nonlocality. As the violation of the CHSH inequality between two spatially separated parties is an important ingredient, I think it would be good that they explain clearly which are the assumptions present in Bell nonlocality that do not appear in their scenario, and what is allowing to abandon them. I am not a fan of repeating it twice or more without giving a useful explanation, it is not magic after all.
- 2) The proof is simple, it can be followed without many problems, but I think a lot of people would appreciate more the paper if the authors could give more intuition about what is happening with their inequalities, how is indefinite causal order related to the violation of the CHSH inequality, or more importantly why definite causal order makes it impossible for Bob and Charlie to violate the CHSH inequality. They make a comment about the first two terms in the inequality being high, but my suggestion is to try to add a bit of "meat" there. This is not crucial to the paper, just a matter of taste.
- 3) I am not sure how common is to use the term "locality" for "parameter independence"? Locality usually has some additional meaning, then the assumption that inputs are not correlated with hidden variables.
- 4) I am a bit suspicious about the name "local-causal inequalities" the authors use, as local causality is basically the synonym for Bell locality if I am not mistaken. They have locality and definite causal order, locality and causality sound too close to local causality and I believe it might furthermore complicate the messy situation with terminology.
- 5) It might be a bit confusing that we have a hidden variable that determines the causal order, and we have local hidden variables determining measurement outputs. Are these in the current paper different things, or is it the same thing, how they are related? I think it would be easier for the general readership if this thing is clarified, it might seem quite confusing.

Finally, congratulations to the authors on a nice result, and overall well-written paper.

Response to reviewers

- We would like to sincerely thank both Reviewers for their careful consideration and useful suggestions. In the below, we will first reply to each of the Reviewers' comments. We will then give some more information about other changes we have made to the manuscript that we felt were necessary.

Reviewer #1

The submitted manuscript considers a central and fundamental question concerning causality in quantum theory, "Is it possible to have Device-independent certification of indefinite causal order within quantum systems?". This paper provides substantial progress by presenting a fair scenario where the quantum switch allows device-independent certification of indefinite causal order.

Apart from an important detail discussed below, the paper is well written, clear, and direct. Additionally, the methods and ideas are presented in a very accessible manner.

I expect this work to have a positive impact and I strongly recommend for publication (see however the comments below).

Technical comments/suggestions:

For this work, the authors have used the software PANDA to obtain all the facets of the LC polytope in this dichotomic given scenario. They have also organized the inequalities in 219 equivalence classes. However, they do not provide their code, nor their data. It would be useful and practical for the community to have a file with all the vertices and inequalities. Not providing their code for sorting the inequalities and data (vertices and inequalities of LC) go against the transparency of science and the open source philosophy.

The authors should provide the code/data used in this work in some online repository.

- We appreciate this suggestion and have included a link to the code and resulting vertex files.

In the abstract, we can read:

"Despite previous evidence suggesting a negative answer"

and in the discussion section, we read:

"It has long been believed that the indefinite causal order in the quantum switch does not admit a device-independent certification [5, 26, 28, 32, 33, 37, 40, 42, 49]. In contrast to this belief, the present result shows that the conjunction of free randomness, definite causal order, and locality (in terms of parameter independence) is inconsistent with correlations observed in the presence of the quantum switch"

In this current form, the submitted manuscript suggests that previous works may have wrong claims, or that they were mere evidence that device-independent certification of indefinite causality with the quantum switch is not possible. However, this is not true. There are rigorous and proper mathematical proofs of this fact. The key point here is that the definition of device-independent certification used in previous works is not the same used in the submitted manuscript. The submitted manuscript also required an extra hypothesis, the hypothesis of locality. I understand that this is a fair hypothesis which is well justified by the submitted manuscript. Also, the expression "device-independence" may be used in a broader and rather vague manner. However, it is important to re-write some sentences of the submitted manuscript to avoid this misconception. It should be clear for the readers that the definition of

device-independent used to prove previous theorems is not the same definition of device-independent used in the submitted manuscript.

- The two quoted sentences in the abstract and discussion have been modified to clarify the relation to previous results.
- We would like to note that although our certification of indefinite causal order relies on the additional Locality assumption, we believe this does not change the notion of device-independence when compared to previous results on causal inequalities. This is because the result is still formulated fully in terms of probability distributions on operational input and output variables; in particular, the Locality condition is formulated within this device-independent framework. In fact, the Locality condition that our result depends on is an integral part of many other device-independent results, such as, notably, device-independent cryptography.
- That said, we agree it is important to be clear about the presence of this additional (device-independent) assumption of Locality. On the other hand, note that it can be seen as natural to include the Locality condition in the Definite Causal Order condition, and that this has previously been done in the context of causal inequalities (see the Discussion in the manuscript). In this sense, our result can be taken to provide a device-independent certification of indefinite causal order without any further assumptions (apart from Free Randomness); however, for full transparency we have chosen to separate off the Locality condition.

I'd say that the main merit of this work is to identify how to make use of LC scenario.

With that in mind, Eqs.27,28,29,30 are very enlightening. They summarize the scenario in a clear and direct manner. I'd suggest the authors to include it in the main text. Also, I'd suggest the authors to add a picture to illustrate LC1 and LC2, this would make the manuscript more accessible for readers from a different area. Fig.3 is very enlightening to clarify the quantum switch strategy, but Fig.1 does not fully clarify the scenario.

- Thank you for this suggestion. We have replaced Figure 1 with an alternative figure illustrating LC1 and LC2 and the definition of LC (now called CLF) as the convex hull of those polytopes. We have also described the intuition behind this definition in the main text and included the Equations that were previously in Methods as suggested.

Line 109: Although the term PR box is well known to the Bell NL community, the authors should define it or at to cite a reference here.

- Thanks for flagging this oversight. A citation and brief definition have been added in the revision.

Line 125: It could be useful for the readers if the authors explicitly states that Alices' measurement choice does not depend on their input x_i for instance, something like "Independently of the classical input x_i , Alice i measures the incoming target system T in the computational basis,"

- Thank you for the suggestion. This point has been clarified in the revised manuscript.

Line 263: It is not clear what is the overlap of the submitted manuscript with Ref.61 (Ref.61 has 474 pages!). The authors could write a couple of sentences on that.

- We have expanded the Note Added with more information on the overlap between the two works.

Reviewer #2

The submitted manuscript considers an important problem of certifying indefinite causal order. When it comes to experimental certification scientists could hope to certify a process with indefinite causal order that can be actually implemented in a laboratory. If this is to be done in the black box scenario, the obvious choice would be the violation of causal inequality. However, the big problem for the community is that all indefinite causal order processes that are implementable, most notably the quantum switch, for now, cannot violate causal inequalities (akin to entangled states which could not violate any Bell inequalities). Some previous results exist in semi-device-independent certification of indefinite causal order.

Here, the authors for the first time show a way for device-independent certification of a process that has indefinite causal order but does not violate any causal inequality. They develop an inequality, satisfied by all processes with definite causal order, and show that quantum switch can be used to provide a violation. This is done by introducing two additional parties, one that is spacelike separated from the main parties undergoing quantum switch and another one in the future cone of the "main" parties. It is these two additional parties that violate CHSH inequality only if the main parties are subject to indefinite causal order.

I think that the paper is very interesting, the result is important and could be motivation for further research in a similar direction. I quite like the idea of definite causal order preventing CHSH violation and I think this submission merits publication in Nature Communications after clarifying a few things.

I have a few suggestions/questions:

1) The authors several times claim that there are fewer assumptions than in Bell nonlocality. As the violation of the CHSH inequality between two spatially separated parties is an important ingredient, I think it would be good that they explain clearly which are the assumptions present in Bell nonlocality that do not appear in their scenario, and what is allowing to abandon them. I am not a fan of repeating it twice or more without giving a useful explanation, it is not magic after all.

- Thank you for the suggestion. We have made several improvements to reflect this:
 - We mention in the Introduction that our Locality notion corresponds to parameter independence, and that we do not assume outcome independence, which Bell locality relies on.
 - In the first subsection of Results, we have added a paragraph explaining the relation of our Locality assumption to Bell locality, stressing again that we do not assume outcome independence.
 - We have changed the Proof Sketch of the Theorem in such a way that we believe explains more clearly why it is not necessary for us to assume outcome independence (nor determinism).

2) The proof is simple, it can be followed without many problems, but I think a lot of people would appreciate more the paper if the authors could give more intuition about what is happening with their inequalities, how is indefinite causal order related to the violation of the CHSH inequality, or more importantly why definite causal order makes it impossible for Bob and

Charlie to violate the CHSH inequality. They make a comment about the first two terms in the inequality being high, but my suggestion is to try to add a bit of "meat" there. This is not crucial to the paper, just a matter of taste.

- Thanks for this suggestion. We have improved the Proof Sketch to make it clearer about the relation to Bell's theorem and CHSH inequalities. We have also added some text to the Discussion that reiterates the intuition behind the argument.

3) I am not sure how common is to use the term "locality" for "parameter independence"? Locality usually has some additional meaning, then the assumption that inputs are not correlated with hidden variables.

- We acknowledge that there are some contradictory naming conventions within research on Bell's and related theorems, and that locality is sometimes used to refer to parameter independence (PI) + outcome independence (OI), and sometimes to only PI (notably by Jarrett (Noûs, 1984). We have chosen to use the term Locality for the following reasons.
 - We did not want to use the term parameter independence as the name of our assumption, as this is a more technical term that is only known to a relatively small audience. Moreover, the term parameter independence suggests a notion that is too broad: for example, the signalling constraints arising from the 'timelike' causal orders (between, say, Alice 1, Alice 2, and Charlie) are also mathematically expressed as 'independence of parameters'. This is incompatible with the fact that we want our locality notion to refer only to the consequences of the spacelike separation between Bob and the rest.
 - In a recent result that partly inspired ours, Bong et al. (Nature Physics 16, 2020) use the term locality for a condition closely related to ours (see Discussion). We hope this choice of terminology emphasises the link between these two results.
 - Partly to clear up terminology confusion, and partly to clarify the meaning of the assumptions and their consequences in general, we have revised our formulation of the three assumptions. In this new formulation, Locality is strictly speaking no longer synonymous with parameter independence: in particular, rather than being a statement about probabilities, it is a statement about partial orders (which implies parameter independence when taken together with the Free Choice assumption).

4) I am a bit suspicious about the name "local-causal inequalities" the authors use, as local causality is basically the synonym for Bell locality if I am not mistaken. They have locality and definite causal order, locality and causality sound too close to local causality and I believe it might furthermore complicate the messy situation with terminology.

- We were aware of this problem and have now come up with a term that we think is better: CLF inequalities, short for Causal Order, Locality, and Free Choice. Apart from avoiding confusion, this term also acknowledges all of our three assumptions.

5) It might be a bit confusing that we have a hidden variable that determines the causal order, and we have local hidden variables determining measurement outputs. Are these in the current paper different things, or is it the same thing, how they are related? I think it would be easier for the general readership if this thing is clarified, it might seem quite confusing.

- There is only one hidden variable at play in our article. This variable, denoted λ , is assumed to take one of finitely many values on each of the experimental trials performed by the parties. Each value of λ determines a probability distribution over the outcomes of the parties given their settings, obtained by conditioning on that value of the hidden variable. In this sense, the hidden variable determines *probability distributions over measurement outcomes*.
- However, these probability distributions are not assumed to be deterministic; thus, the hidden variable does not necessarily determine particular measurement outcomes that will be observed with certainty: it does not *predict* the outcomes.
- The main role of this same hidden variable is that it determines the causal order. What this means is that it takes two values, corresponding to the two possible causal orders, and that the probability distribution of measurement outcomes given settings obtained by conditioning on each value of λ is compatible with the respective causal order.
- A potential source of confusion is that the argument of the paper (in particular Theorem 1) works by noting that under certain conditions on the data (viz. the first two inequality terms adding up to 1, met by the quantum switch setup), the fact that λ determines the causal order *implies* that it is also perfectly correlated to one of the measurement outcomes (b for $y=0$). In other words, it predicts that outcome (but not necessarily the others).
- We hope that this clarifies the situation. We have improved the formulation of the assumptions and the subsequent discussion in the first subsection and hope that this makes it more insightful. We have also changed the Proof sketch of Theorem 1 to be more clear about the relation with Bell's theorem, in particular to emphasise that the correlation between b for $y=0$ is deduced from Definite Causal Order, rather than postulated.

Finally, congratulations to the authors on a nice result, and overall well-written paper.

- Thank you!

Other changes

Here we discuss any changes that have not already been discussed in the above replies.

- As mentioned above, we have revised the formulation of our three assumptions. This required a restructuring of much of the first section of Results. We have chosen to do this for the following reasons.
 - The new formulation is more precise, both mathematically and physically. For example, it is clearer about what the role of each of the assumptions is separately. Moreover, it specifies more explicitly what is meant by causal order (namely a partial order; a concept that precedes correlations) and how Free Choice imposes constraints on correlations, given a particular causal order.
 - In the original manuscript, the latter role was played by Locality, which was a poor choice as Locality should only refer to 'spacelike' aspects of causality. Indeed, precisely those spacelike aspects are the main novelty of this work.
 - It allows for a more rigorous derivation of the independence relations constituting the polytope. The previous derivation might have seemed a bit ad hoc.
 - The terminology confusion regarding 'Locality' discussed above.

- We have largely restructured the last section of Results ('More CLF inequalities'). Upon some thought we decided that much of the material presented in that section was too specific and technical to include in the main text. We have moved that material to a new section in Methods, while the main text only presents the table of inequalities and summarises the results. We have also updated the table to include a new inequality (iv), which was discovered after submission of the first version of this manuscript. We believe this new inequality should be included as it illustrates the adaptability of the proof strategy of our main inequality in Theorem 1, and might therefore inspire future research; and also because it exhibits some advantages with respect to the other inequalities, as discussed in the revised manuscript. (Nevertheless, it should not replace Equation (9) as the main inequality of our paper, because of its higher level of complexity.) Intuition for this new inequality is discussed in the revised last section of Results, paralleling the intuition for the first inequality given earlier in the text, while it is rigorously proven in Methods.
- We have added a reference to Bong et al. (Nature Physics 16, 2020), which partly inspired our work, in the Discussion.
- Throughout the manuscript we have made minor and mainly linguistic corrections.

REVIEWER COMMENTS

Reviewer #1 (Remarks to the Author):

In this resubmission, the authors have considered the points I've raised in my previous report. As I wrote in my previous report, I'm very much towards publication. However, there are two technical points I should make.

1 - I'd say that using the word locality for the purpose used in this work raises some confusion and ambiguity. The word locality has several different meanings in different context in quantum info, and outside the quantum info community, I don't see any gain in sticking with terminology. Specially because it's not equivalent to Bell locality. As written by the authors:

"Moreover, note that our notion of Locality is a weak one. It, along with Free Choice, entails what is known as parameter independence in the context of Bell's theorem [45], and in our case leads, for example, to Equation (6)."

In the ultimate case, if the definition is mathematically precise, the authors are, of course, free to use the terminology they prefer.

However, currently, the definition of locality is still not precise.

In the current version of the manuscript, the definition of locality, is made only with text (lacking mathematical and clarity). The definition is::

"Locality: Bob is causally unrelated to Alice 1, Alice 2 and Charlie, regardless of the value of λ . This can, for example, be motivated by placing Bob at spacelike separation from the other parties."

Does this text correspond to Eq.6 ?

When the formal and mathematical equation appears (Eq.6,7,8), it is not clear what is meant by locality. From my understanding of their text definition, locality is precisely non-signalling, that is, Bob's marginals do not depend on the inputs of Alice 1, Alice 2, and Bob. Also, the marginals from Alice 1, Alice 2, and Bob do not depend on the inputs of Alice. Hence, locality seems to be the NS condition described in Eq.6. However, they seem to need the Free choice assumption to obtain the NS condition of Eq.6, since they wrote:

"Moreover, note that our notion of Locality is a weak one. It, along with Free Choice, entails what is known as parameter independence in the context of Bell's theorem [45], and in our case leads, for example, to Equation (6)."

The authors could write a formal definition of locality and Free Choice, and how they are combined to obtain the NS condition of Eq.6. (For their theorem to hold, they just need NS, so why not assume NS directly? That's not clear to me...)

2 - Before Thm.1 the authors write:

"Moreover, note that our notion of Locality is a weak one. It, along with Free Choice, entails what is known as parameter independence in the context of Bell's theorem [45], and in our case leads, for example, to Equation (6)."

Equation 6 is also widely known in the community as "non-signalling".

Review round #2

Reviewer #1

- Thank you very much for going through the revised manuscript again and providing useful comments which have been helpful in improving the manuscript.
- In the below, we reply to each of your points individually. Let us however start by noting an important fact about the structure of our assumptions. Namely, the first two assumptions are assertions about the causal order, which is considered as an a priori concept described by a partial order \prec_λ , and which does not in itself make statements about correlations between stochastic variables. Only the third assumption ('Free Interventions') talks about correlations: it puts constraints on correlations based on the partial order \prec_λ introduced in the first two assumptions.
- Thus, in particular, the statistical independence conditions often referred to as 'no-signalling' constraints require this third assumption and do not follow from the first two assumptions alone. This is how we have chosen to formulate our assumptions; we will give motivation for this choice below.
- These remarks holds for both the first revision and the current, new revision, even though we have slightly restructured the assumptions in this new revision and made them more formal. We hope that this makes the remarks above immediately clear to the reader.

In this resubmission, the authors have considered the points I've raised in my previous report. As I wrote in my previous report, I'm very much towards publication. However, there are two technical points I should make.

1 - I'd say that using the word locality for the purpose used in this work raises some confusion and ambiguity. The word locality has several different meanings in different context in quantum info, and outside the quantum info community, I don't see any gain in sticking with terminology. Specially because it's not equivalent to Bell locality. As written my the authors:

"Moreover, note that our notion of Locality is a weak one. It, along with Free Choice, entails what is known as parameter independence in the context of Bell's theorem [45], and in our case leads, for example, to Equation (6)."

- We agree that Locality is an overloaded term. A slight restructuring of the assumptions has enabled us to change the name from Locality to Relativistic Causality.
- First of all, in the text leading up to the assumptions (beginning of results) we have been more explicit about the spatiotemporal configuration that we consider the experiment to take place in (Charlie after Alice; Bob spacelike).
- The Definite Causal Order assumption then asserts that there is a definite causal order, while the Relativistic Causality assumption roughly speaking asserts that this causal order is 'compatible' with the spatiotemporal structure of the experiment. This captures both what might be called 'no retrocausation' (which was previously included in Definite Causal Order) and 'no superluminal causal influence' (which one might call locality, as we did in the first revision, but which is not Bell locality).
- We hope that this structure makes it clearer what the role of the second assumption is - namely, it puts restrictions on the partial order defined in the first assumption, and does not relate to correlations.

- We believe that in our context, the term Relativistic Causality does not suffer from the same ambiguities as Locality does, and the higher level of rigour with which we formulated the assumption should make its meaning clear.

In the ultimate case, if the definition is mathematically precise, the authors are, of course, free to use the terminology they prefer.

However, currently, the definition of locality is still not precise.

In the current version of the manuscript, the definition of locality, is made only with text (lacking mathematical and clarity). The definition is:

"Locality: Bob is causally unrelated to Alice 1, Alice 2 and Charlie, regardless of the value of λ . This can, for example, be motivated by placing Bob at spacelike separation from the other parties."

Does this text correspond to Eq.6 ?

- We have improved our articulation of the formal content of the first two assumptions. Making the treatment completely formal requires too much space for the main text (e.g. one needs to account for dynamical causal order), so we have made some shortcuts which are justified in a new, completely formal section of Methods and Supplementary Note 2.
- To answer your question, the Locality assumption on its own did *not* correspond to Equation (6), nor does the new Relativistic Causality assumption. Equation (6) (now (2)) is about correlations and to derive it (more precisely, to derive that the conditional distributions $p(\cdot | \cdot, \lambda)$ are members of the NS polytope defined in this equation), one needs Free Interventions in addition to Relativistic Causality.
- To stress the fact that the first two assumptions are not about correlations, we now introduce the random variables and correlations between them only *after* formulating the first two assumptions.

When the formal and mathematical equation appears (Eq.6,7,8), it is not clear what is meant by locality. From my understanding of their text definition, locality is precisely non-signalling, that is, Bob's marginals do not depend on the inputs of Alice 1, Alice 2, and Bob. Also, the marginals from Alice 1, Alice 2, and Bob do not depend on the inputs of Alice. Hence, locality seems to be the NS condition described in Eq.6. However, they seem to need the Free choice assumption to obtain the NS condition of Eq.6, since they wrote:

"Moreover, note that our notion of Locality is a weak one. It, along with Free Choice, entails what is known as parameter independence in the context of Bell's theorem [45], and in our case leads, for example, to Equation (6)."

- You are correct that Eq. (6) can only be derived from the conjunction of Relativistic Causality and Free Interventions (see the previous bullet point). While Relativistic Causality is an assumption about an priori notion of causality, Free Interventions allow us to translate it to a constraint on correlations (sorry to keep repeating this but it's an important aspect to note). We hope that the restructured assumptions and their higher level of rigour will make these points clear to readers.

The authors could write a formal definition of locality and Free Choice, and how they are combined to obtain the NS condition of Eq.6. (For their theorem to hold, they just need NS, so why not assume NS directly? That's not clear to me...)

- As suggested, we have made the formulation of the assumptions more formal. We have also added a completely rigorous Methods section and Supplementary Note.
- It might be confusing why we haven't chosen to formalise Definite Causal Order (DCO) and Relativistic Causality (RC) in a way that directly includes their consequences for

correlations in terms of statistical independences (e.g. parameter independence). There are several reasons for this; here we give two examples.

- Firstly, while it is relatively clear how $p(\cdot | \cdot | \lambda) \in \text{NS}$ might intuitively follow from RC, the case for $p(\cdot | \cdot | \lambda) \in \text{CRF}_\lambda$ is more complicated. Since joint independence $a_1 b \perp\!\!\!\perp x_2$ does not follow from the individual independences $a_1 \perp\!\!\!\perp x_2$ and $b \perp\!\!\!\perp x_2$ (which would be implied by DCO and RC individually), it'd be difficult to argue for $a_1 b \perp\!\!\!\perp x_2$ if we used the alternative formulation of the assumptions described above. In our choice of formulation, on the other hand, Free Interventions captures the correlation consequences of both DCO and RC in one go and therefore implies $a_1 b \perp\!\!\!\perp x_2$ (conditioned on $\lambda=1$).
- Second, the assumption that measurement settings are freely chosen is crucial to the physical motivation for assuming the mathematical condition of parameter independence, even though this is not often explicitly noted in discussions of Bell's theorem and the like. Indeed, if settings aren't freely chosen then there could be common causes between (say) x_1 and b , making them correlated while being perfectly consistent with relativity theory.
 - (Note that Free Interventions in the main text consists of two parts: part (i) is akin to 'free choice'/'measurement independence' in Bell's theorem while part (ii) is akin to 'parameter independence'. Note that both in fact follow from a single natural assumption which can be found in the new Supplementary Note.)
- For these reasons we have decided to capture all assumptions that involve correlations in the Free Interventions assumption. As a bonus, we get a nice parallel with causal inference literature, where ('free') interventions are often seen as providing the crucial stepping stone between causation and correlation.

2 - Before Thm.1 the authors write:

"Moreover, note that our notion of Locality is a weak one. It, along with Free Choice, entails what is known as parameter independence in the context of Bell's theorem [45], and in our case leads, for example, to Equation (6)."

Equation 6 is also widely known in the community as "non-signalling".

- The relevance of the NS polytope in our argument is that $p(\cdot | \cdot | \lambda) \in \text{NS}$ for each λ , i.e. there are independences *conditioned on the hidden variable*. These conditional independences are equivalent to parameter independence, while non-signalling usually refers to unconditional independences. Thus, while the polytope of Eq. (6) is indeed the no-signalling polytope, its relevance to the story of the paper lies in parameter independence.
- Thanks again for your useful comments. For ease of reference, here is summary of the most important changes we made:
 - On the first page of Results, we restructured the assumptions; in particular, it is stressed that 'Relativistic Causality' puts constraints on the partial order \prec_λ and is not related to correlations, while Free Interventions is.
 - We added a Methods section and Supplementary Note 2 containing a more formal treatment than that in the main text. It deals with dynamical causal order in a way based on previous literature.

REVIEWERS' COMMENTS

Reviewer #1 (Remarks to the Author):

This resubmission offers great improvement and clarity when comparing to the first one. I find the text easier to follow, also, the ambiguities and issues from previous version have been properly addressed. In my view, this work constitutes very positive progress for the field and it is ready to published in Nature Communications.

Review round #3

Reviewer #1

This resubmission offers great improvement and clarity when comparing to the first one. I find the text easier to follow, also, the ambiguities and issues from previous version have been properly addressed. In my view, this work constitutes very positive progress for the field and it is ready to published in Nature Communications.

- We would like to thank the reviewer once more for their time and effort in helping us improve this manuscript.